# Laser-Induced Surface Modification on Wollastonite-Tricalcium Phosphate and Magnesium Oxide-Magnesium Stabilized Zirconia Eutectics for Bone Restoring Applications

**Shunheng Wang [1,2], Daniel Sola [3,4] and Jose I. Peña [1,*]**

1 Instituto de Nanociencia y Materiales de Aragón, Universidad de Zaragoza-CSIC, 50018 Zaragoza, Spain
2 School of Materials Science and Engineering, Tiangong University, Tianjin 300387, China
3 Laboratorio de Óptica, Centro de Investigación en Óptica y Nanofísica, Campus Espinardo, Universidad de Murcia, 30100 Murcia, Spain
4 Aragonese Foundation for Research and Development (ARAID), Government of Aragon, 50018 Zaragoza, Spain
* Correspondence: jipena@unizar.es

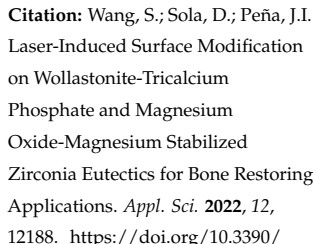



**Featured Application: Bioceramic materials for bone restoration.**

**Abstract:** An adaptation of the laser floating zone technique is used to modify the surface properties of ceramics with interest for biomedical applications. This new method is based upon the surface remelting of ceramic rods by using laser radiation, and its versatility is demonstrated in the surface structuring of two different eutectic composites with potential application as bone substitutes. Firstly, directionally eutectic rods of wollastonite (W)–tricalcium phosphate (TCP) and magnesium oxide (MgO)–magnesium stabilized zirconia (MgSZ) were grown by the laser floating zone technique. In the case of W-TCP eutectics, materials with crystalline, glass–ceramic, or vitreous microstructure could be obtained as the growth rate was increased. In the other case, a material made up of magnesium oxide and magnesium stabilized zirconia phases arranged in fibrillar or lamellar geometry was obtained. At higher solidification rates, the rupture of the growth front gave rise to the organization of the phases in the form of colonies or cells. The laser zone remelting technique was used to remove defects and to refine the microstructure of the directionally solidified eutectic surfaces as well as to cover MgO–MgSZ rods with W–TCP glass in the eutectic composition. The study provides a promising technique that can tailor the surface properties and functionality of bone repair materials. The products' properties and challenges in preparation procedures are discussed.

**Keywords:** laser floating zone; directionally solidified ceramic eutectics; magnesium oxide; zirconium oxide; wollastonite; tricalcium phosphate; bioactive glasses

## 1. Introduction

One of the major challenges in modern regenerative medicine is to develop biomaterials that are both mechanically strong and bioactive. The mechanical properties are necessary to prevent any structural failure during normal activity of the patient or during handling the implant in the operation. Bioactivity is desirable for the bone repair process to dominate fibrous repair. Usually, in the field of bioceramics, these properties are antagonist [1].

Eutectic ceramics grown from the melt have a great potential as structural and functional materials due to their high resistance, low density, and synergic combination of phases. At present, the ceramic materials used in bone implants are polycrystalline and usually fabricated by conventional sintering method, requiring high temperatures to achieve high densification. The presence of pores and defects at grain boundaries reduce the mechanical behavior of these materials. Directionally solidified eutectics have excellent

mechanical properties inherent to the full density, homogeneous microstructure, reduced interphase spacing, and large surface area of clean, strong interfaces. From the biological point of view, among the eutectic ceramics we can find some that have a bioactive behavior and others are bioinert.

De Aza et al. [2] have shown that glasses and ceramics in the systems containing CaO-$P_2O_5$-$SiO_2$ can be designed to optimize biological and mechanical response. They have reported in the phase diagram of the system wollastonite ($CaO \cdot SiO_2$)–tricalcium phosphate ($3CaO \cdot P_2O_5$) an invariant point at 1402 °C and a eutectic composition of 80%mol of wollastonite (W) and 20%mol of tricalcium phosphate (TCP). Further studies concluded that ceramics and glasses with this composition can be used in bone repair applications. Such ceramics, known as Bioeutectic, have the property of developing a porous hydroxyapatite-like structure in simulated biological conditions by the dissolution of the wollastonite phase and transformation of the tricalcium phosphate phase [3]. The role of Si ions on the biomineralization process has been widely reported. Si ions substitution in CaP ceramics and glasses has effects on the differentiation, proliferation, and collagen synthesis of osteoblasts and facilitates precipitation of hydroxyapatite (HAp) on the bioceramic surface by increasing the solubility of the material through the creation of crystalline defects with substitution for $PO_4^{3-}$ for $SiO_4^{4-}$ generating a more electronegative surface [4].

Alumina and zirconia-based eutectics have high strength, wear resistance, and fracture toughness but poor bioactivity [5]. Zirconia-based materials are especially interesting in orthopaedics (bone screws, knee implants) and maxillofacial applications and also as dental material because of their aesthetics characteristics (translucency and colour). For example, tetragonal zirconia polycrystal with three percent of yttria (3Y-TZP) has been widely used in dental applications. However, hydrothermal degradation leads to microcracking, grain pull-out, and finally surface roughening [6]. More stable zirconia can be achieved in 3Y-TZP by adding a small amount of alumina, which is homogeneously distributed in the zirconia matrix [7]. It is considered that the use of yttria as a dopant is the cause of ageing of zirconia because of the creation of oxygen vacancies by yttrium ions, thus inducing the diffusion of hydroxyl groups in the lattice and consequently giving rise to stress corrosion type mechanism. Coatings to be used for thermal barriers based on $ZrO_2$ stabilized by CaO and MgO have been the purpose of intensive research aiming at producing protective layers and to be applied to promote the adhesion in Ti-ceramic dental restoration as reported by E. Marcelli et al. [8]. Magnesia- and calcia-doped zirconia ceramics have been object of little consideration in bone restoration because of their lower toughness. On the contrary, they do not show ageing and the use of eutectic structures could open a door for zirconia ceramics with improved properties.

L. Grima et al. reported the generation of a porous scaffold by the rapid dissolution of MgO phases when a multiphase bioceramic was in contact with simulated body fluid (SBF). The release of Mg ions to the fluid inhibited the apatite formation, and a bioactive porous matrix with interconnected channels was formed [9].

In this study, rods of W-TCP and MgO–MgSZ, both in the eutectic composition, have been grown from their melts by laser-assisted directional solidification. The effect of the solidification rate on the microstructure, the mechanical properties, and the dissolution behavior in simulated body fluid (SBF) of these eutectics are studied. For the first time, the porosity formation due to the MgO dissolution after immersion in SBF of the MgO–MgSZ eutectic is reported.

A new method based on a modification of the laser zonal fusion technique (LFZ) is presented. It consists of laser remelting of the eutectic rods to change their surface properties. The application of this method results in the formation of a glass layer on the W–TCP eutectic rods and a refinement of the surface microstructure of the MgO–MgSZ eutectic rods. The effect of the formation of these layers is discussed in terms of improvement of the mechanical properties and dissolution behavior.

After soaking the remelted rod in SBF, a porous layer of MgSZ is obtained. The effect of the release of magnesium ions and the appearance of pores on the bioactivity of the

samples is discussed. Using this new technique, a functional glass coating has been created on the MgO–MgSZ eutectic rods to improve their limited bioactivity.

## 2. Materials and Methods

Sample preparation required the following reagents: zirconium oxide (Aldrich, 99%, St. Louis, MI, USA), magnesium oxide (Aldrich, 99.9% USA), calcium silicate (Aldrich, 99% USA), tricalcium phosphate (Carlo Erba Reagenti, Cornaredo, Italy, Analytical Grade), ethanol (PanReac, 99.8%, Barcelona, Spain), polyvinyl butyral (PVB, Sigma-Aldrich USA), and Beycostat C213 (CECA, Chevagnes, France).

A powder mixture was prepared according to the reported eutectic composition (53 MgO; 47 $ZrO_2$ in mol%) by using commercial powders of $ZrO_2$ and MgO [10]. These ceramic powders were used to fabricate cylindrical precursors by isostatic pressing for 3 min at 200 MPa. The rods of about 3 mm in diameter and 7 cm long were subsequently sintered at 1500 °C during 12 h.

The eutectic composite of wollastonite (W) and tricalcium phosphate (TCP) was prepared by isostatic pressing of the powder mixture followed by a sintering process at 1200 °C for 12 h, with resulting precursor rods of diameter and length ranging 3 mm and 50–100 mm, respectively. Eutectic composition, 80 $CaSiO_3$; 20 $Ca_3(PO_4)_2$ in mol%, and melting point were obtained from the phase diagram of calcium silicate (CS)–Tricalcium phosphate (TCP) reported by De Aza et al. [11].

Directionally solidified eutectic samples of MgO–MgSZ (magnesium stabilized zirconia) and W–TCP were grown with the LFZ technique as described elsewhere [12]. As the heating source a continuous wave $CO_2$ laser was utilized, and the growth process was carried out in air. Densification stages were applied to eliminate the porosity of the precursor rod using pulling rates of 200 mm/h and a counter rotation in the liquid zone of 50 rpm. To avoid bubbles in the eutectic samples, the growing process was performed downwards. Specifically, for the MgO–MgSZ samples, rates of 50, 100, 300, and 750 mm/h were used without rotation to avoid phase segregation, and 20, 100, and 150 mm/h with 50 rpm counter rotation for W-TCP. The final diameter for the whole samples ranged from 1–2 mm. The W-TCP samples were annealed for 2 h at 600 °C to relieve internal residual stresses. The surface remelting process of the eutectic rods was performed by pulling the rod downwards with rotation to homogenize the heating generated by the laser beam focused on the sample surface.

The W–TCP bioceramic coatings on MgO–MgSZ eutectic rods were prepared by dip coating technique. The substrate rods were dipped and withdrawn several times from the ceramic solution at a fixed speed. The composition of the solution was in wt%: 42.37 W-TCP; 44.49 ethanol; 0.42 Beycostat C213 (as the dispersant agent); 12.72 PVB (as the binder). The thickness layer was controlled by adjusting the concentration of the solution and number of dips. Ten dips were used at a speed of 3 mm/s. No crack formation or spallation were observed on the coating surface after the sintering treatment at 1200 °C during 12 h, below the melting point of 1402 °C. This treatment was necessary to improve the densification and adherence of the coating to the substrate. The coated cylinders were superficially melted using the LFZ technique. The power was adjusted to melt the coating with minimum dilution with the substrate. A laser power of 20 W was enough to melt a W–TCP layer of about 200 μm.

Characterization of the microstructure and analyses of the composition were performed by using Field Emission Scanning Electron Microscopy, FESEM (FESEM, model Carl Zeiss MERLIN), and the Energy Dispersive X-ray Spectroscopy (EDS) detector coupled to the microscope. Analyses were carried out in polished samples in both transversal and longitudinal configurations. In the case of crystalline W–TCP eutectics, etching was used to reveal the structure. Due the poor contrast between both phases, the samples were etched with dilute acetic acid to remove the TCP phase. The volume fractions of the phases were obtained by analysis of SEM images taken on polished samples without etching. The crystalline nature of the W–TCP eutectic was identified by X-ray diffraction (XRD) (model RU 300, RIGAKU)

working at 40 kV and 80 mA, using $K_\alpha$ radiation (1.5418 Å). The scanning was carried out between 5° and 70° (2Theta) in 0.03° steps, counting for 1 s per step.

Indentation tests were performed to assess the Vickers Hardness of the samples (Matsuzawa, MXT70). The procedure was realized following the ASTM C1327-99 Standard. An indentation load of 4.9 N and a holding time of 15 s were used for the measurements, which were carried out in polished cross-sections for no less than 10 valid tests. Determination of the fracture toughness was evaluated based on the both the mark and the crack lengths obtained from the indentation tests.

Three-point flexural tests were performed in 1 mm diameter as-grown samples to assess the bending strength (Instron testing machine, model 5565). Loading span and crosshead speed were set at 10 mm and 30 μm/min, respectively. Tests were realized at room temperature. To determine the mean values and standard deviation, five valid tests were performed. It was observed that samples showed a linear load-displacement trend. The maximal load obtained during the test allowed determining the flexural strength, according to the standard beam theory.

For the estimation of the in vitro bioactivity of the MgO–MgSZ composite, the test proposed by Kokubo was used. Discs were cut form the rod samples and maintained for four weeks in simulated body fluid (SBF). The ion concentration was observed to be nearly equal to that of the body plasma as a SBF environment ($Na^+$ 142.0, $K^+$ 5.0, $Mg^{2+}$ 1.5, $Ca^{2+}$ 2.5, $Cl^-$ 148.8, $HCO^{3-}$ 4.2, and $PO_4^{2-}$ 1.0 mM), and the solution was buffered at pH 7.2. The temperature of the samples was kept at human body temperature, 37 °C, by using a stove (Memmert GmbH, model 100–800, Schwabach, Germany).

## 3. Results and Discussion

### 3.1. W-TCP Eutectic

A remarkable feature of the W–TCP eutectic grown by the laser floating zone technique is the possibility of obtaining glass, glass–ceramic, or crystalline microstructures depending on the growth conditions, which enables them to be used for desired application because of their different biological activity and mechanical behavior. Crystalline materials can be obtained at low growth rates of 20 mm/h. Its appearance is opaque, white, and with a tendency to crack due to high thermal gradients generated upon solidification. Vitreous cylinders can be obtained at speeds above 150 mm/h, adjusting the power of the laser beam to avoid the nucleation of crystallites in the glass. To prevent the precipitation of crystalline phases, the melt must be cooled sufficiently fast defining a minimum cooling rate. At solidification rates between 75 and 150 mm/h phosphate crystalline phases nucleate in a glassy matrix of calcium silicate producing a glass–ceramic material. Scanning electron microscope images of the samples grown at 20 mm/h are presented in Figure 1. The volume fraction of W is 0.61 and the phases are arranged in colonies attached to each other and formed by alternating lamellae.

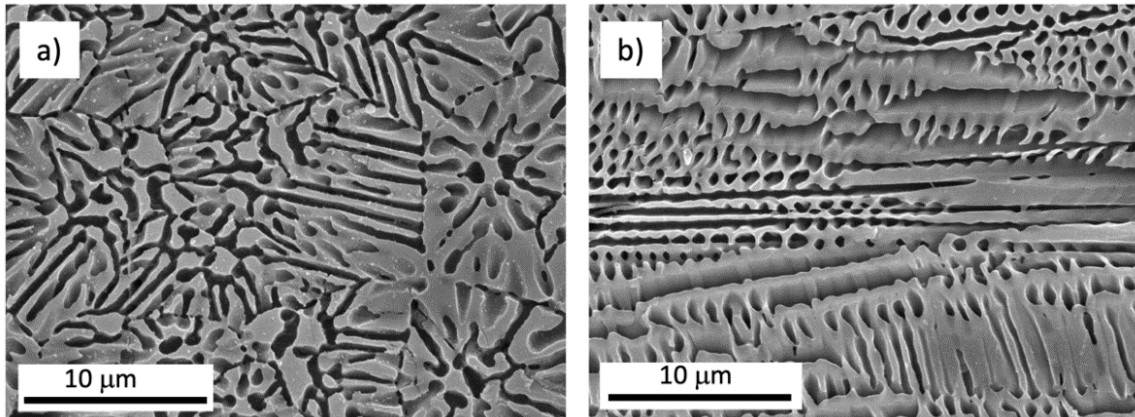

**Figure 1.** SEM Images of a transversal (**a**) and longitudinal (**b**) cross-section of a W-TCP rod grown by LFZ at 20 mm/h. The sample surfaces were etched with dilute acetic acid to remove the TCP phase.

Figure 2 presents the microstructure of a eutectic rod grown at 100 mm/h. The images were taken at the edge of the rod, and the phases are evenly distributed throughout the sample. Due to the greater thermal gradient generated with respect to the center of the sample, the size of the phases is smaller at the edge, but the morphology is the same. EDS analysis showed that the darker phase (52.3 vol%) corresponded to calcium silicate with a small amount of phosphorus in solution (Ca/Si = 1, Ca/P = 7.38) while the lighter phase was silicon calcium phosphate (Ca/P = 2.21; Ca/Si = 3.33 and Ca/P + Si = 1.33).

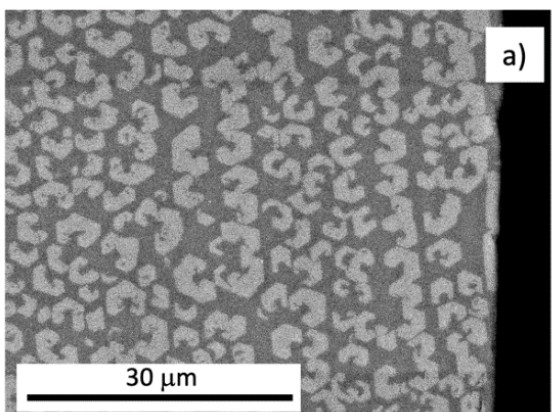 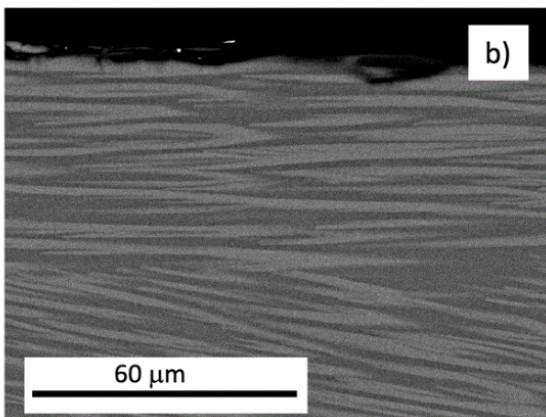

**Figure 2.** SEM Images of a transversal (**a**) and longitudinal (**b**) cross section of a W–TCP rod grown by LFZ at 100 mm/h. Lighter phase corresponds to silicon calcium phosphate and darker phase to calcium silicate.

The results of qualitative XRD analysis of the crystalline, glass–ceramic, and glass samples are shown in Figure 3. The XRD powder patterns show the presence of apatite and pseudo-wollastonite (ps-W) for the crystalline sample and apatite for the glass–ceramic sample. No diffraction peaks were detected in the eutectic glass.

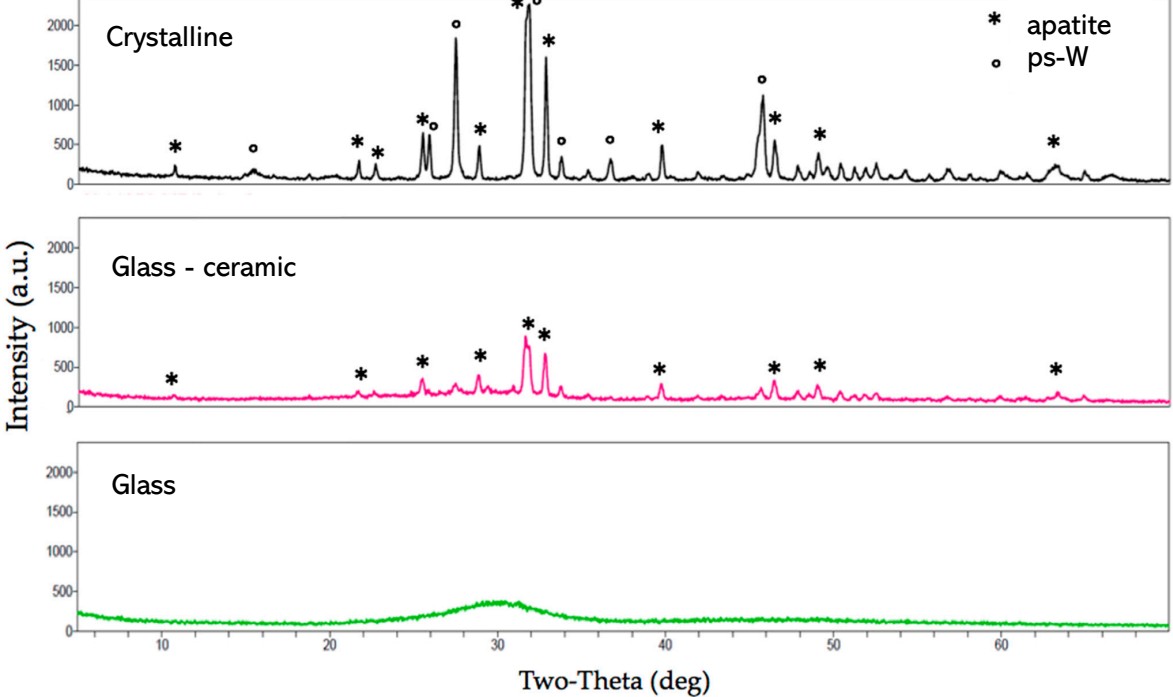

**Figure 3.** XRD patterns of the samples grown at 20 mm/h (crystalline), 100 mm/h (glass–ceramic), and 150 mm/h (glass).

We can conclude that at high solidification rates, the eutectic material has an amorphous structure, at lower speeds an apatite crystalline phase precipitates, and at slower speeds two crystalline phases are formed, apatite and pseudo-wollastonite. The glass–ceramic microstructure with crystalline phases of apatite in a glassy phase of calcium silicate is in good concordance with the devitrification studies of wollastonite–tricalcium phosphate eutectic glass carried out by M. Magallanes-Perdomo et al. [13]. These authors reported that the devitrification starts with the precipitation of apatite crystals from the glass followed by the crystallization of wollastonite.

Dimensionless parameter $\gamma = T_x/(T_g + T_l)$ firstly proposed by Lu and Liu [14] was used an indicator of the glass-forming ability of the system, where $T_x$ is the crystallization temperature, $T_g$ the glass transition temperature, and $T_l$ the liquidus temperature. The critical cooling rate ($R_c$) for glass formation is correlated to the $\gamma$ parameter by means of the following expression:

$$R_c = R_o \exp \left[ (-\ln R_o)\gamma/\gamma_o, \right. \tag{1}$$

where the constants $R_o$ and $\gamma_o$, for oxide glasses, are $8 \times 10^{27}$ K/s and 0.421, respectively.

The devitrification and crystallization process of W–TCP eutectic glass begins at 870 °C with the crystallization of a Ca-deficient apatite phase [13]. Accounting that the glass transition temperature for this glass was estimated at 790 °C, the calculated $\gamma$ parameter and critical cooling rate $R_c$ were found to be 0.414 and 3 K/s, respectively. This rate corresponds to a solidification rate of about 100 mm/h in the Laser Floating Zone (LFZ) technique for an experimental axial thermal gradient of $10^5$ K/m, in good accordance with the growth rate of 150 mm/h reported by Pardo et al. for the fabrication of a W–TCP eutectic glass rod of 3 mm in diameter [15]. It is also similar to other critical cooling rates of oxides given in the literature [16].

This eutectic glass, produced by the laser floating zone technique, is transparent as seen in Figure 4a. In this figure, the precursor rod located at the top of the figure and the glass resulting from the manufacturing process at the bottom can be observed. As mentioned before, the fabrication process is carried out downwards. Figure 4b corresponds to a W–TCP rod, grown at two different rates. The translucent part, on the left, corresponds to a glass–ceramic composite and the opaque part, on the right, is totally crystalline; they are separated by a glass portion. We can conclude that it is possible to obtain a range of microstructures from the eutectic composition, with different crystalline phases, through appropriate design of solidification conditions.

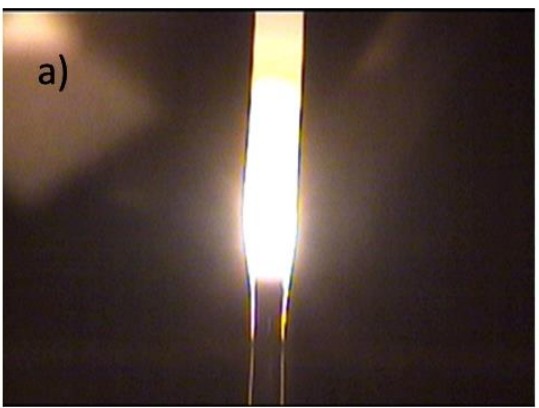
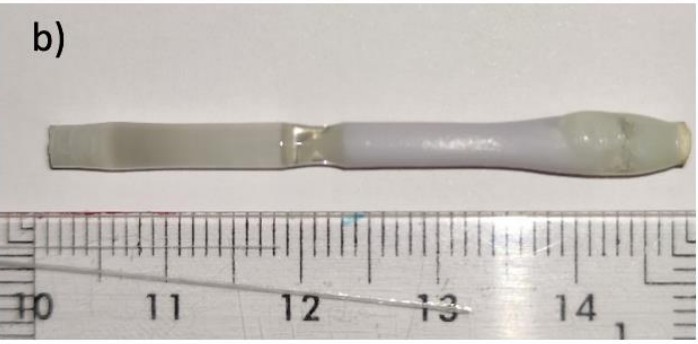

**Figure 4.** Growth by LFZ of a glass rod of W–TCP with the eutectic composition (**a**). Eutectic W–TCP sample (**b**) grown at two different conditions (100 mm/h left part and 20 mm/h right part).

According to the glass composition, its bioactivity can be predicted by the $N_c$ factor that corresponds to the number of bridging oxygen bonds per silicon atom. The formula expressing this factor is [17]:

$$N_c = \frac{4[SiO_2] - 2[M_2O + MO] + 6[P_2O_5]}{[SiO_2]} \tag{2}$$

For this glass $N_c = 2$, glasses of $N_c > 2.6$ are likely not bioactive due to their resistance to dissolution.

The composition of this eutectic (33.33 $SiO_2$-58.33 CaO-8.33 $P_2O_5$) corresponds to an "invert" glass, as firstly proposed by Trapp and Stevels [18], where the modifier content (Ca) is larger than the former content (Si + P), which contains less bridging oxygen resulting in isolated parts and shorter chains. The term "invert" was proposed since the traditional network forming oxides $SiO_2$ and $P_2O_5$ form continuous molecular/ionic networks in normal conditions. Nevertheless, if the network modifying oxides, like in this glass, are in majority on the molar basis, the glasses are structurally inverted compared to conventional glasses. This structural inversion is reflected in the properties of the glasses as reported by I. Lebecq et al. [19]. Their work permitted to show that the phosphorus-containing invert glasses are active, developing the apatite layer in relatively short immersion times (10–16 h) in SBF.

The appearance of different crystalline or glassy phases coexisting in the material controlled by the solidification conditions would allow to make a specific material, with the same composition but different mechanical response or biodegradation rate depending on the application. The crystalline form of this material is composed by two phases: a bioactive phase capable of transforming into hydroxyapatite in contact with the physiological medium and wollastonite, whose dissolution generates a network of interconnected pores. The glassy form shows quick formation of a hydroxyapatite layer on its surface when they are immersed in SBF, due to the release of calcium and silicon ions during the first stage of immersion and the formation, in a second step, of an amorphous layer rich in silicon that provides nucleation points for the apatite crystals. It is expected that in the glass–ceramic material, the higher solubility rate of the amorphous phase leads to the formation of an interconnected porous structure of residual apatite [20].

Mechanical characteristics of these materials were measured by indentation technique and bending test. The results are given in Table 1.

**Table 1.** Mechanical properties of W–TCP rods with different microstructure.

| Material | HV (GPa) | Toughness (MPa m$^{1/2}$) | Flexural Strength (MPa) |
|---|---|---|---|
| Crystalline | $4.6 \pm 0.45$ | $1.39 \pm 0.3$ | 73.8 |
| Glass ceramic | $5.1 \pm 0.78$ | $0.99 \pm 0.28$ | 82.3 |
| Glass | $4.9 \pm 0.18$ | $1.08 \pm 0.12$ | 310 |

The low resistance of the crystalline rod is due to the presence of cracks caused by the high thermal gradients inherent to the solidification process and the difference in thermal expansion coefficients of both phases ($\alpha_W = 6.5 \times 10^{-6}$ °C$^{-1}$ [21] and $\alpha_{TCP} = 14.2 \times 10^{-6}$ °C$^{-1}$ [22]). The rods containing a glassy phase can adjust in a better way the internal stresses upon solidification and relieve them under annealing because of their plasticity at high temperature. The best results in flexural strength are achieved in the fully glassy rods. Hardness and toughness are less dependent on microstructure, as seen in Table 1.

As the cracks emerge from the surface during the bending test, an increase in the flexure strength has been achieved with a laser surface remelting of the crystalline rods. This treatment generates a superficial glass layer capable of closing the cracks and at the same time can be used to modify the surface bioactivity. Figure 5a shows optical and SEM micrographs corresponding to longitudinal and transverse cross sections of eutectic

rods remelted by laser, at 300 mm/h. A rotation of 200 rpm was applied to the rod to homogenize the distribution of energy provided by the laser on its surface.

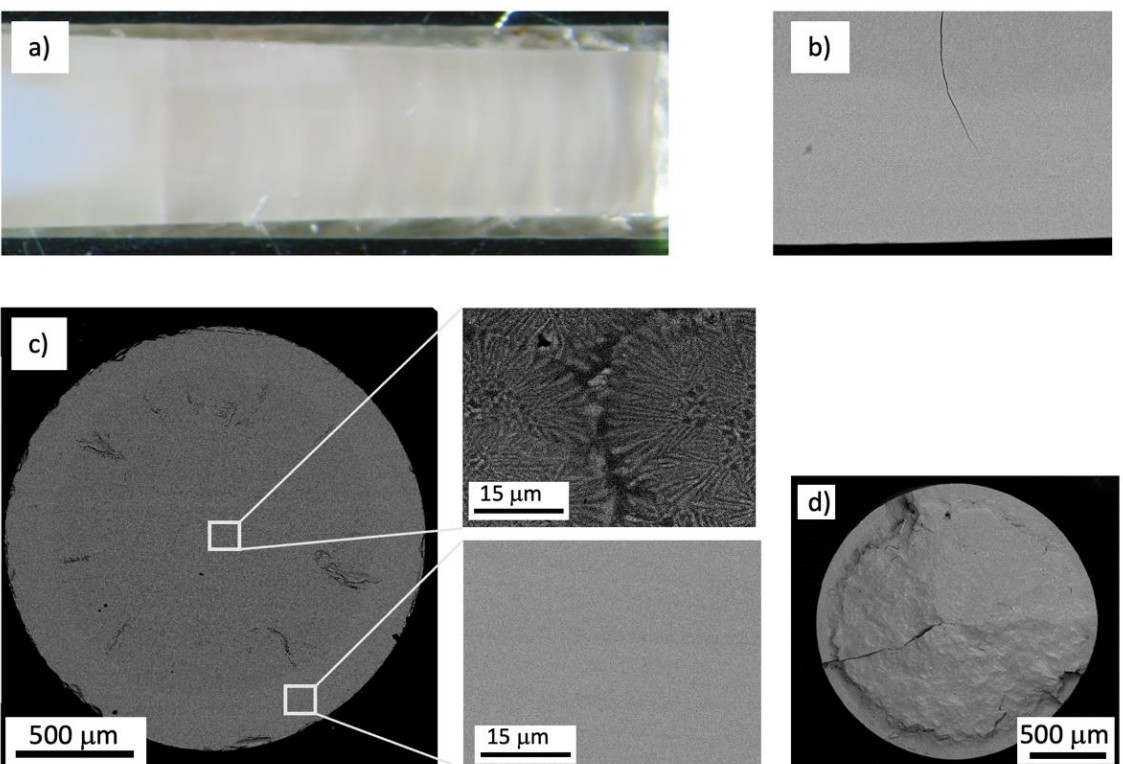

**Figure 5.** Optical image of a longitudinal section of a glazed eutectic rod (**a**). Crack arrest through superficial laser melting (**b**). Cross section of a eutectic rod with multiple flaws (**c**). Details of the microstructure at the center and edge of the rod are shown. Fracture section of a glazed eutectic rod showing the crack causing the failure (**d**).

The flexural strength of the glazed eutectic rod after being annealed was 129 MPa. This value indicates, therefore, that the glass surface formation strengthens the crystalline eutectic rods. The flaws are arrested by the glass layer, as shown in Figure 5b, hindering the rapid crack propagation during the bending test. Figure 5c shows the cross-section of a eutectic rod with multiple flaws with details of the microstructure at the center and edge of the rod.

As ceramic fracture is a process of crack nucleation and propagation, this processing approach could be exploited in many technological applications where ceramics and glasses are used, allowing these materials to be used in a more reliable way. In Figure 4d, the fracture section of a glazed eutectic rod is shown, displaying the crack that caused the failure.

### 3.2. MgO-MgSZ Eutectic

Rods of MgO–MgSZ were obtained by directional solidification of precursor rods prepared according to the eutectic composition of the system $MgO:ZrO_2$ (53:47 in mol%). Figure 6 represents FESEM micrographs of the transverse (a) and longitudinal cross-sections (b) in the region of the center and edge of a sample solidified at 50 mm/h. The eutectic material was composed of two phases, one of magnesium oxide (MgO) and other of magnesium stabilized zirconia ($Mg_{0.2}Zr_{0.8}O_{1.8}$) with a volume fraction of 0.28 and 0.72%, respectively. At this rate the microstructure consisted of grains of thin MgO fibers embedded in a zirconia matrix and oriented in the rod axis, and alternating lamellae oriented in different directions, extended throughout the cross section of sample. The longitudinal section indicated that the phases are mainly oriented perpendicular to the solidification front.

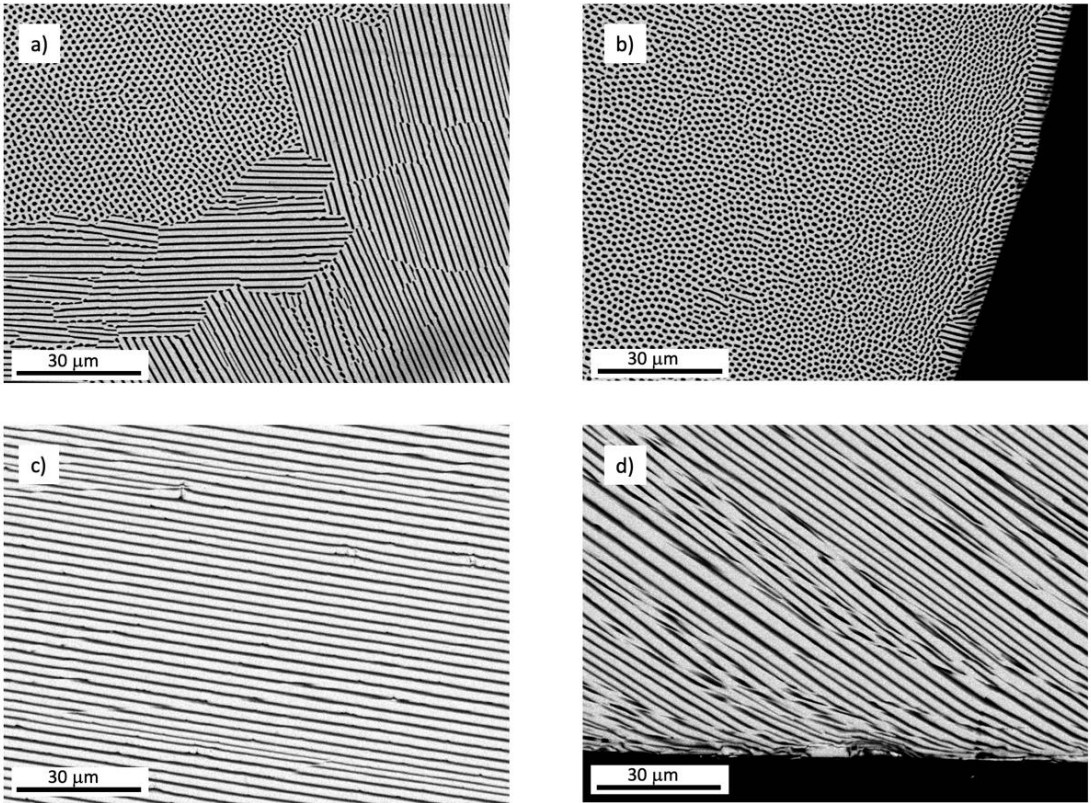

**Figure 6.** FESEM micrographs of the transverse and longitudinal cross-sections of the specimens grown at 50 mm/h in the region of the center (**a**,**c**) and the edge of the sample (**b**,**d**). Dark phase corresponds to magnesium oxide and bright phase to magnesium-stabilized zirconia.

At 150 mm/h, the resulting microstructure was mainly lamellar, as shown in Figure 7a,b. However, at 300 mm/h colonies started to develop in the entire section of the sample, Figure 7c,d, due to the breakdown of planar solid–liquid interface. A total change to cellular structure occurred at solidification rate of 750 mm/h, Figure 7e,f. The cells presented round shape in the center of the sample and were formed by an ordered dispersion of MgO fibers. At the edge, the microstructure was fibrillar (see insert in Figure 7f). The cells were surrounded by a thick intercellular region formed by coarse MgO and MgSZ particles of irregular shape. It is worth noting that at faster solidification rates, the microstructure is mainly fibrillar. However, at slow growth rates, there are grains with a fibrillar phase distribution and others with a lamellar phase arrangement. Fibrous structures are found in eutectics in which the minority phase corresponds to a volume fraction less than 28%. Above this percentage, the phase distribution is laminar, and the MgO volume fraction in this eutectic is close to the transition from fibrillar to lamellar geometries. Thus, very small changes in the solidification process could produce this transition, being more frequent in the center of the cylinder where the gradients are not as high as at the edge. The formation of colonies is explained by the rupture of the flat solidification front due to the high growth rates.

A good agreement of the calculated eutectic spacing $\lambda$ with the Jackson and Hunt theory was observed, for which $\lambda = kV^{-1/2}$, where V is the growth rate and k a constant of proportionality, which was calculated based on the experimental, obtaining a value of 8.24 $\mu m^{3/2}$ $s^{-1/2}$, close to the one reported by Kennard [10] for MgO–MgSZ eutectics ($\sim$7 $\mu m^{3/2}$ $s^{-1/2}$). This constant depends on the individual partition coefficient for each phase ($k_{\alpha}$, $k_{\beta}$) and also on the difference between the two phases (e.g., $k_{\beta}-k_{\alpha}$) and indicates the correlation of phase spacing and growth rate. The constant makes it possible to predict the phase separation for a determined solidification rate under the minimal undercooling conditions.

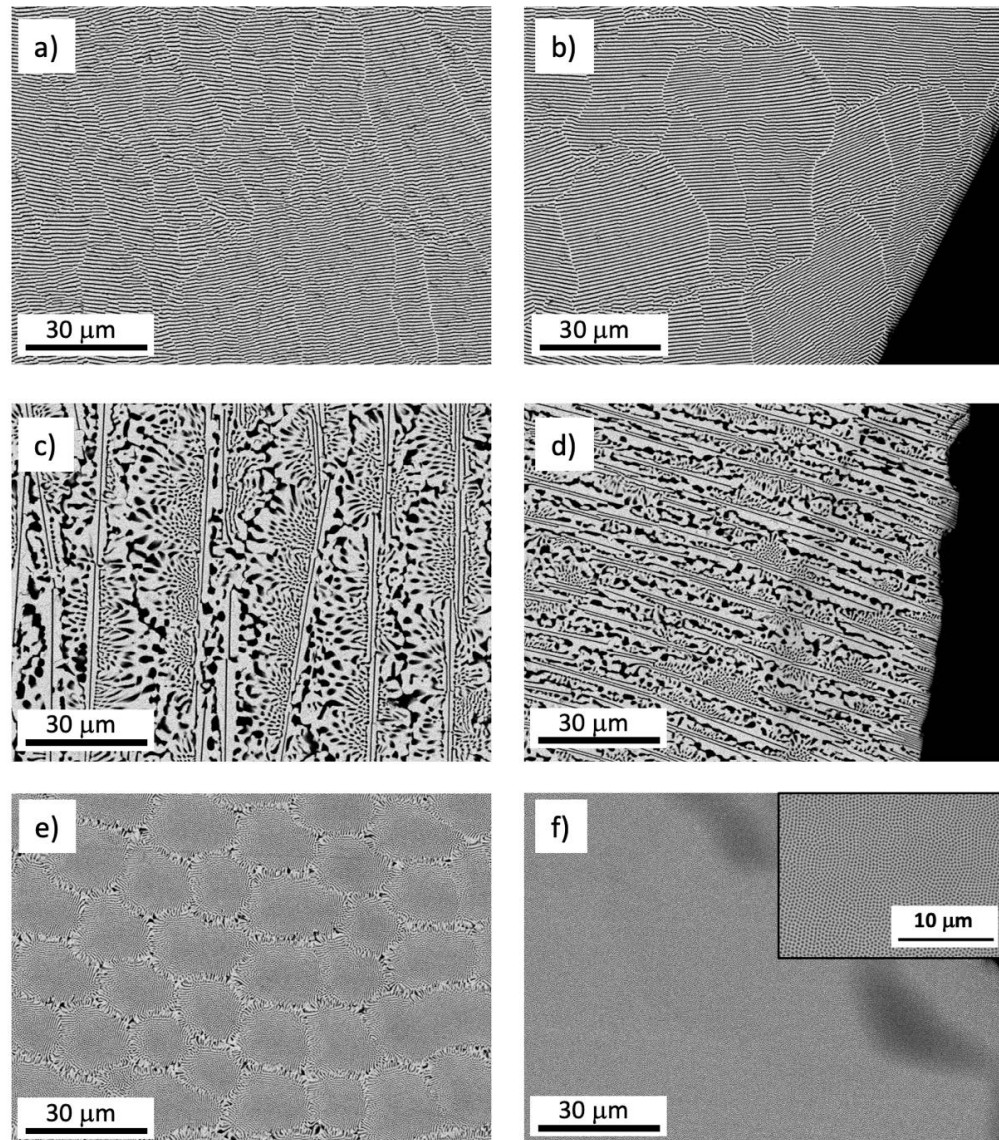

**Figure 7.** FESEM micrographs of transverse cross-sections of samples grown at 150 mm/h (**a**,**b**), 300 mm/h (**c**,**d**), and 750 mm/h (**e**,**f**), in the center and in the edge, respectively.

Hardness and indentation fracture toughness were obtained from Vickers microhardness tests on polished transverse cross sections. The obtained results of 11 GPa and 1.5 MPa m$^{1/2}$, respectively, were coherent with those reported by Sola et al. [23]. A best flexural strength of 900 MPa were obtained in samples grown a 750 mm/h, despite the presence of cells surrounded by a coarse microstructure. However, the edge of the cylinders (more than one hundred of microns in depth) has a homogeneous, fibrillar structure with a very small phase spacing (320 nm). This fact would explain the greater difficulty for the formation of the crack during the bending test.

### 3.3. Surface Structuring of MgO–MgSZ Eutectic by Laser Remelting and Phase Dissolution in SBF

In this section, we explore two approaches of surface structuring MgO–MgSZ eutectic: surface laser remelting and dissolution of one of the eutectic phases by immersion of the material in SBF.

The strength of a eutectic ceramic can be improved by minimizing the structural dimensions, in terms of lamellar or fiber spacing. Laser remelting consists in a surface solidification technique with high temperature gradients that significantly increases the nucleation rates, resulting in refined feature structures and hopefully with improved

mechanical properties. In our case, we used the laser floating zone set up creating a superficial molten zone that is translated along the eutectic rod. The penetration of the laser-treated layer in the rod can be feasibly controlled by adjusting solidification rate by the moving laser radiation, and the laser power.

The fully eutectic microstructures with lamellar and nano-sized morphology can be observed in Figure 8. Their lamellar spacing was about 1 μm in as-solidified eutectic (noted as AS) and 150 nm in the laser remelted layer (noted as LRL), indicating that laser remelting can significantly refine the structure and provides a promising increase in mechanical strength. The lamellar structure in AS is anisotropic and ordered, and in LRL the lamellae were organized in cells oriented in the laser direction. No phase segregation, crack, or porosity were found on the surface and inside of the sample.

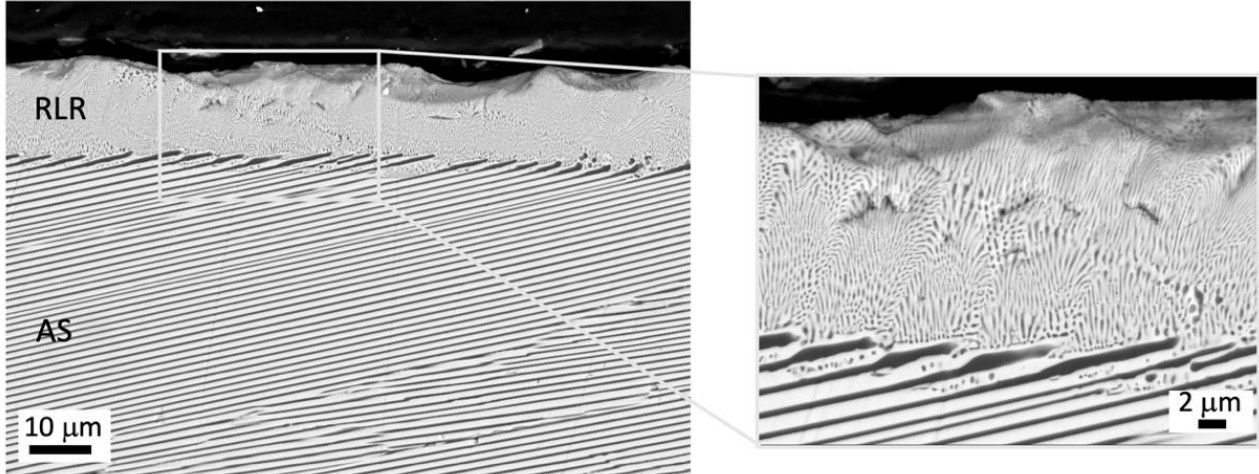

**Figure 8.** Microstructure of the longitudinal cross section of the as-solidified MgO–MgSZ eutectic grown at 50 mm/h and subsequently remelted with laser at 100 mm/h and rotation of 50 rpm.

In Figure 9, a cross-section of a lamellar MgO–MgSZ eutectic rod after soaking in SBF at 37 °C for 4 weeks is shown. The microanalyses performed by EDS confirmed that the composition of the layer in contact with the liquid is $Mg_{0.13}Zr_{0.86}O_{1.73}$, and the composition in the inner zone corresponds to the eutectic MgO–MgSZ. The liquid has penetrated about 80 microns into the sample, dissolving the MgO phase and generating an MgSZ skeleton whose sheets tend to collapse. This last fact hinders the possibility of achieving a surface with a controlled porous topography. Better control can be achieved with the use of eutectic samples that have been subsequently remelted. The fast dissolution of MgO phase in a physiological environment has already been described by L. Grima et al. [9] and used to generate in situ porous scaffolds. In this case, the amount of released magnesium ions corresponds to the difference in magnesium composition in zones 1 and 2 of Figure 8.

The cylinders that have been remelted on their surface showed the same behavior, but the smaller surface pore size allowed the topography to be maintained once the MgO was dissolved in the medium, as shown in Figure 10. In practice, the control of the melt volume is carried out by regulating the laser power and the translation speed of the cylinder, which also determines the phase spacing. Figure 10 corresponds to the cross section of a cylinder treated with a laser at three different powers moving the rod at 100 mm/h. The depth of the resulting layers was 2, 8.5, and 14 μm for laser powers of 15, 25, and 30 W, respectively. The phase size of the treated layer determines the topography and the size and distribution of pores on the resulting surface after immersion in SBF.

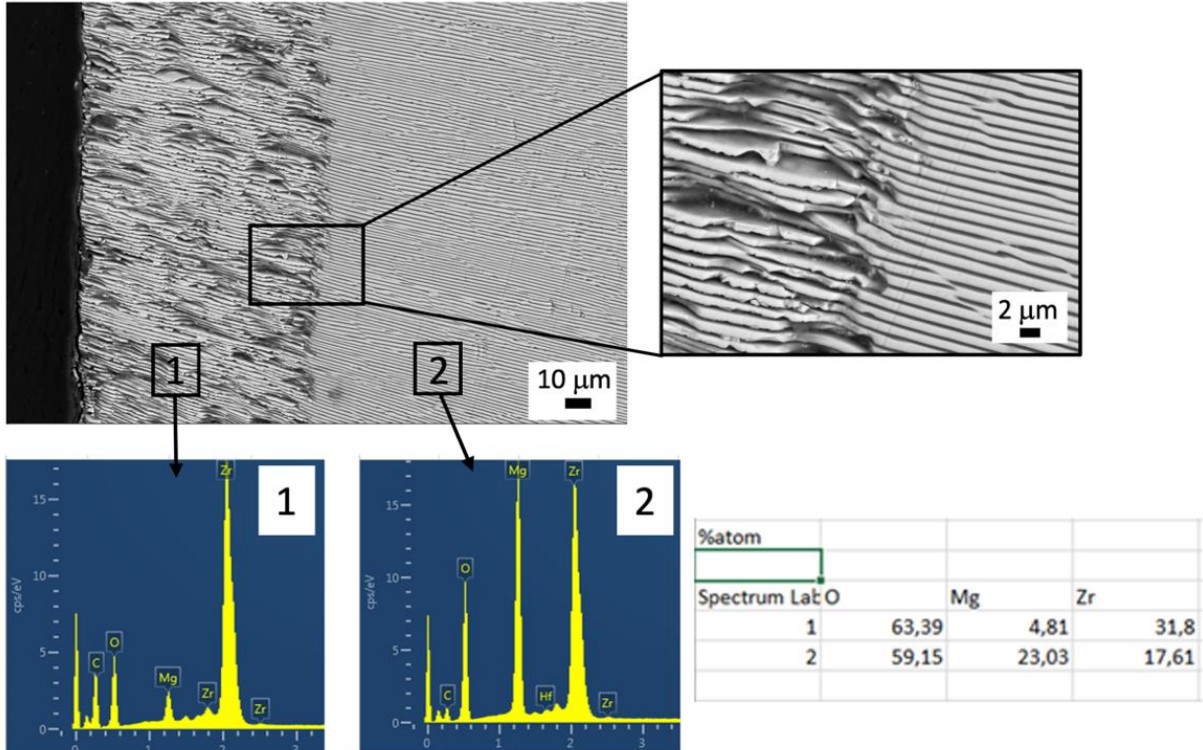

**Figure 9.** FESEM image of a cross section of MgO–MgSZ eutectic after soaking in SBF for 4 weeks. Right: interface detail. Below: Chemical composition obtained by EDS of a region in the porous layer (**1**) and in the eutectic zone (**2**).

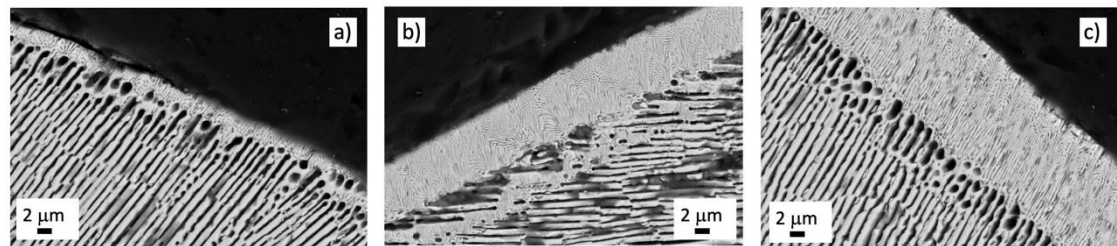

**Figure 10.** FESEM images of eutectic samples of MgO–MgSZ superficially remelted with different laser power (increasing power from left to right: 15, 25 and 30 W, (**a**–**c**), respectively) and kept in SBF for 4 weeks at 37 °C.

Except for the dissolution of MgO with the consequent generation of a porosity of about 28%, no change in the surface of the sample was observed. The release of Mg ions and the inert character of zirconia do not favor the formation of a HAp layer on the surface of the sample. However, the poor HAp crystallization on the surface of zirconia does not exclude the possibility of protein and cell adhesion promoting stable bonding between implant and the living bone. In the interaction with cells and tissues, the topography of the bioceramic surface can influence the absorption of proteins from body fluids influencing cell attachment [24]. By avoiding the collapse of the zirconia sheets and with the formation of small pores on the surface, in the range of 0.2 μm, the wettability of the biomaterial's surface could be modulated and with it its biological response as reported by L. Hao and J. Lawrence [25].

Both the surface topography and the surface free energy are the key factors which mostly influence on the wettability of a surface. The surface of the zirconia, which presents hydrophilic features, will present higher hydrophilic characteristics after surface roughening. Biological fluids prefer to spread on a hydrophilic surface, providing more area for

protein adsorption and interaction with cell receptors. However, it is also possible to make the surface hydrophobic by laser treatment as described by Yilbas, who found that laser-treated surfaces composed of micro/nano grooves improved the surface hydrophobicity of yttria-stabilized zirconia [26]. However, in this type of surface, the contact area between cell and surface is reduced as air bubbles may be confined in between, resulting in a lower adsorption of proteins. To provide more information about this behavior, a comprehensive study on the effect of the topography on the wettability of the surface is being carried out in our samples.

This method could be considered a soft way to obtain a surface with microporosity avoiding any mechanical or thermal cracking like those caused by other machining techniques such as sandblasting or laser ablation. It is known that thermal damage generated by laser texturing or laser ablation reduces the mechanical strength of the material [27]. Etching techniques do not apply stresses but could make some adverse chemical changes on the surface of the material and therefore should be taken into consideration when designing surface textures for certain biological functions.

Microporosity is an important parameter, as it determines the dissolution properties, protein adsorption and surface area of bioceramics. On the other hand, free $Mg^{2+}$ ions released from the material are considered as positive influence on various cell reactions in the human body such as proliferation, osteogenic differentiation, and matrix formation [28].

*3.4. W-TCP Coatings on MgO-MgSZ Rods*

In the previous section, physical modifications in eutectic MgO–MgSZ cylinders have been analyzed by modifying the surface topography as well as chemical changes associated with the dissolution of one of the phases. In this section, another well-established chemical technique for surface functionalization is considered—the coating of the substrate surface to create a specific chemical environment that offers a favorable biological response.

A layer of W–TCP ceramic in the eutectic composition was placed by dip coating on the surface of a MgO–MgSZ eutectic rod, sintered, and subsequently melted by means of the modified LFZ technique to integrate the coating on the eutectic rod.

The power of the laser and the rod movement were adjusted to melt the entire layer with a minimum dilution with the substrate. Deliberate conditions were used to obtain a bioactive glass layer on the fully dense body inside, with macro-size pores that could favor tissue integration. In this case, air was used as growth atmosphere, and a sufficiently high solidification rate and layer thickness to prevent the escape of all the bubbles formed in the liquid pool.

Figure 11 shows a longitudinal (a) and transversal (b) cross-section of a W–TCP glass-coated MgO–MgSZ eutectic rod. The coating showed a clean interface suggesting good interfacial strength, as shown in Figure 11c. Accounting that the sample was not preheated, there were no visible cracks or delamination phenomena at the coating/eutectic interface, favored by the cylindrical symmetry of heat flow. It can be observed that small dendrites of MgO–MgSZ were distributed within the darker bioactive phase, close to the interface.

From a mechanical behavior point of view, the hardness and toughness increase from the coating to the substrate. The Young's moduli of substrate and coating are very different, $E_{MgO-MgSZ} = 285$ GPa [23] and $E_{w-TCP}$ is estimated in 110 GPa, considering the Young's modulus of the wollastonite, a tricalcium phosphate, 130 GPa [29], and 72 GPa [30], respectively. The Young's modulus estimated for the coating is closed to the measured in other multiphase ceramics obtained from the melt [31]. The load transfer plays an important role in the success of an implant, and this is favored when the elastic properties are similar. Optimizing the composition, structure, and microporosity of the coating, the Young's modulus can approach the one of the bone tissues (between 12 GPa and 28 GPa for the cortical bone [32]).

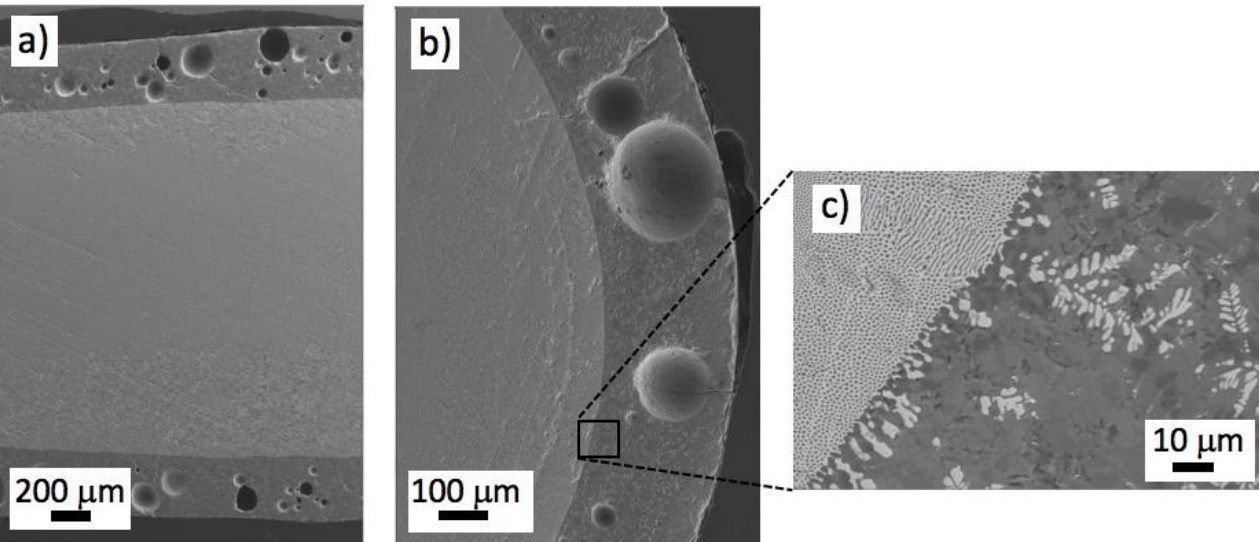

**Figure 11.** Cross section images of deposited bioactive coating—eutectic composite. Longitudinal (**a**) and transversal (**b**) section of the coated rod. As there is not enough time for pores to escape when the laser scanning speed is relatively high and the layer thick enough, those pores aggregate to form small cavities onto the bioactive layer. Eutectic substrate/bioactive coating interface (**c**).

The bioactive layer can be tailored to match a specific tissue healing rate by changing its structure (glass or crystalline) through modification of growth conditions, as described in Section 3.1. It is also possible to add certain elements (Li, Mg, Sr, Cu, Co, Mn, B, Si) to the coating composition that can be released, playing a role in angiogenesis, osteogenesis, and osteoconduction [33,34].

The presence of zirconia dendrites in the coating increases when the composition of the substrate is deviated to the zirconia-rich side. Usually, the microstructure of one off-eutectic composition system consists of coarse primary crystals embedded by eutectic constituent. Figure 12a shows the microstructure of an off-eutectic MgO–MgSZ. The primary phase of MgSZ (bright phase) and the eutectic constituent can be seen clearly. The globular shape of zirconia phase shows a weak-faceted growth. Figure 12b corresponds to the transversal cross-section of an off-eutectic rod covered by a solidified layer of W–TCP. A detail of the coating and interface is presented in Figure 12c. The coating was 115 μm thick and was formed by a bright phase of $Ca_{0,17}Mg_{0,11}Zr_{0,72}O_{1,60}$ and a dark phase with composition Ca/Si = 3.58 and Ca/P = 2.48, close to the theoretical W–TCP eutectic of Ca/Si = 3 and Ca/P = 2.25.

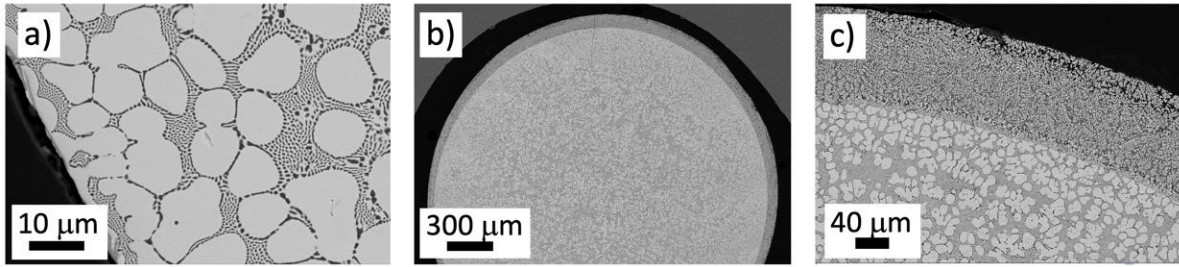

**Figure 12.** FESEM transversal cross-section image showing the microstructure of a directionally solidified off—eutectic ceramic in the MgO–ZrO$_2$ system with excess of ZrO$_2$ (**a**). W–TCP coated rod (**b**). Close-up view of the coating-off eutectic interface (**c**). The dendritic structures seen at the interface and coating were CaMgSZ.

The in vitro behavior of the W-TCP bioeutectic ceramics and glasses has been studied by P. Velásquez et al. [35]. Both materials have the ability of forming a HAp layer in SBF. It is known that bioactive glasses bond with bone more rapidly than its ceramic counterparts and their dissolution products stimulate beneficial response from the body [36]. However, the biodegradable nature of glass can cause the bioactive coating to dissolve over time. Therefore, new compositions should be explored to optimize dissolution and bone integration of the coatings.

## 4. Conclusions

- Directionally solidified eutectics in the W–TCP and MgO–MSZ systems with good comprehensive properties and controlled microstructure have been grown by the LFZ technique. Microstructure characteristics at different solidification rate were investigated.
- W–TCP eutectic composition. Microstructure was strongly dependent on processing rate, obtaining crystalline eutectics, glass–ceramics, or glasses when increased the growth rate. At slow rate of 20 mm/h phases of apatite and pseudo-wollastonite were observed. At moderate rates of 100 mm/h crystalline phases of apatite in a glass matrix of calcium silicate were formed. At rates above 200 mm/h the eutectic rods grew amorphous. Crystalline cylinders were subjected to a superficial laser melting process, generating a glassy layer that improved their flexural strength, from 73.8 MPa to 129 MPa.
- MgO–MgSZ eutectic composition. The microstructure changed from fibrillar/lamellar to colonies and cells with increasing growth rate in the range from 50 to 750 mm/h. Correspondingly, the phase spacing decreased gradually. The relationship between eutectic phase spacing with solidification rate can be summarized as $\lambda = 8.24 V^{-1/2}$. Hardness and the fracture toughness of 11 GPa and 1.5 MPam$^{1/2}$, respectively, were obtained for rods grown at 750 mm/h. Flexural strength of 900 MPa was obtained.
- A simple and useful strategy to prepare rapidly solidified MgO–MgSZ nanoeutectic ceramic rods by laser surface remelting is proposed. After laser remelting, the lamellar spacing was refined sharply, without the need of high temperature preheating to avoid crack generation.
- After soaking in SBF, the MgO phase of the eutectic dissolves, leaving a zirconia skeleton on the surface in contact with the liquid. Pores of around 0.2 µm with approximately the same size as the structural elements (zirconia phase) are homogeneously distributed on the ceramic surface.
- The limited bioactivity of the MgO–MgSZ eutectic can be improved by the use of W–TCP eutectic as a bioactive cladding. The combination of dip coating and laser surface melting was able to meet the requirements in terms of coating thickness, homogeneity, and adhesion to the substrate. No coating spallation from the substrate was observed that could result in adverse clinical response.

**Author Contributions:** Methodology, all authors; validation, all authors; investigation, all authors; writing—original draft preparation, J.I.P.; writing—review, all authors; funding acquisition, all authors. All authors have read and agreed to the published version of the manuscript.

**Funding:** This research was funded by the Departamento de Ciencia, Universidad y Sociedad del Conocimiento del Gobierno de Aragón through the financial support to the Research Group T02-20R. S.W. wants to acknowledge the support of the China Scholarship Council during his sabbatical stay in Zaragoza. D.S. thanks the PIT2 program of the University of Murcia's own research plan.

**Acknowledgments:** Authors acknowledge the use of Servicio de Microscopia Electrónica (Servicios de Apoyo a la Investigación), Universidad de Zaragoza.

**Conflicts of Interest:** The authors declare no conflict of interest. The funders had no role in the design of the study; in the collection, analyses, or interpretation of data; in the writing of the manuscript; or in the decision to publish the results.

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
