# Peer review of "Laser-Induced Surface Modification on Wollastonite-Tricalcium Phosphate and Magnesium Oxide-Magnesium Stabilized Zirconia Eutectics for Bone Restoring Applications"

_applsci, doi:10.3390/app122312188_

Round 1
Reviewer 1 Report
This study named “Surface modification and functionalization of MgO-MgSZ eutectic ceramics for bone restoring applications” investigates eutectic composites with different composition and directionally solidified by using the laser floating zone technique at different processing rates to obtain microstructures with different domain sizes. While the work is interesting, I could not overcome the sense. There are methodological issues and lack of interpretation that need greater clarification and several major and minor issues that need to be addressed before the manuscript can be recommended for publication. For instance, I did not understand that the authors elaborated 3 kind of materials and worked on the surface structuring by laser remelting and phase dissolution until the section 3.3…
ABSTRACT
Abbreviations, and acronyms should be defined the first time they are used.
In the tittle, the authors used “functionalization of MgO-MgSZ eutectic ceramics” which is not clearly described in the abstract. The goal of the paper needs to be explained with a red line to understand for instance why 3 types of materials was elaborated,…
Abstract should explain to the general reader why the research was done, what was found and why the results are important. The final sentence should outline the most important results, the main conclusions of the study.
INTRODUCTION
Not a really smooth introduction with no clear message. I encourage to seek the main message, but also the information necessary to transmit a key message. The introduction should provide sufficient background information to make the article intelligible with sufficient context and with links or comparison with the materials of interest in this study. What are the goals of the study?
The novelty of the work wasn't clearly mentioned in this section. Authors must consider reorganizing the introduction to have a more concrete understanding of what your paper will focus on and distinguish their work from other published research that can be found in the literature.
Full words have to be written instead of abbreviations as YSTZ yttria stabilised tetragonal zirconia (YSTZ), same for 3Y-TZP
Lines 56 and 59 3Y-TCP instead of 3Y-TZP are used.
Line 105 and 107 the authors may use bioactive glass named RKKP bioglaze® (RKKP®)
MATERIALS AND METHODS
All commercial precursors used (powders and solvents: CaSiO3, ethanol; Beycostat….) with details provider and purity needs to be in the first paragraph. If synthesize the protocols need to be described. Full words have to be written instead of abbreviations : psW
In this section, it seems that 3 kind of rods samples are elaborated MgO–MgSZ, W–TCP and W–TCP bioceramic coatings on eutectic rods what is not clear in the abstract and the introduction. In “W–TCP bioceramic coatings on eutectic rods” which rods are coated?
it would be beneficial to put lines from 151 to 154 in the results section. The authors should use a correct unit for “1402 ºC”
RESULTS AND DISCUSSION
3.1. Bioactive W-TCP eutectic composite
In the tittle of 3.1. the word “composite” appears for the first time and should be mentioned earlier in the introduction and materials.
Lines from 184 to 196 : the paragraph should be moved in the Introduction section
Could the authors provide XRD analysis to define the nature of phases and their crystallinity or with amorphous behavior at 20, 100 and 150mm/h?
Could the authors explain why : “The sample surfaces were etched with dilute acetic acid to remove the TCP phase”? And if it is part of the elaboration that could be relocated in the section MATERIALS AND METHODS
If TCP are removed, could the authors provide the nature of the 0.39% volume fraction of phases?
For the eutectic rod grown at 100 mm/h, the sample surfaces were also etched with dilute acetic acid to remove the TCP phase?
Could the authors explain the difference of composition in the sample and why the sample are grown at two different rates, could they add the EDS results?
The significance of the experimental findings is not clear. Is there an interest to obtain 3 different composition and structure?
Line 266 “The samples were previously annealed for 2 hours at 600 ºC to relieve internal residual stresses” all details as this one are beneficial in the section MATERIALS AND METHODS
The remelting process could be explained and described with all the parameters in the section MATERIALS AND METHODS
The authors have to mention the conditions at which they obtained the results and add them in the figure caption
Is the test to estimate the in vitro bioactivity could be done for this sample?
3.2. MgO-MgO eutectic
The authors should correct the tittle of this section
Could the authors provide XRD analysis to define the nature of phases and their crystallinity?
In Figure 5, could the authors describe the two different phases with caption
Could the authors explain the difference of composition in the sample, could they add the EDS results?
Could the authors compare and discuss the mechanical properties of the W-TCP composite and MgO-MgZ
3.3. Surface structuring of MgO-MgSZ eutectic by laser remelting and phase dissolution
The section is not described earlier in the manuscript…
Line 369 Could the authors edit the word lamellae
Could the authors determine the amount or % of the Mg ions released
Could the authors explain the impact of presence of porosity or loose of Mg ions?
Could the authors provide a porosity analysis of the samples?
3.4. W-TCP coatings on MgO-MgSZ rods
Could the authors provide the in vitro behavior of the W-TCP coatings on MgO-MgSZ materials?
Could the authors describe and discuss more in this section?....
References
The authors have to consistently follow the same citation style when citing an article
Author Response
Dear reviewer, thank you for your suggestions and for the time spent to analize this manuscript. We have tried to correct all the points that you have indicated. The corrections are listed below:
ABSTRACT
Abbreviations, and acronyms should be defined the first time they are used.
In the tittle, the authors used “functionalization of MgO-MgSZ eutectic ceramics” which is not clearly described in the abstract. The goal of the paper needs to be explained with a red line to understand for instance why 3 types of materials was elaborated,…
Abstract should explain to the general reader why the research was done, what was found and why the results are important. The final sentence should outline the most important results, the main conclusions of the study.
Thank you, the title and abstract has been changed according to your suggestions
INTRODUCTION
Not a really smooth introduction with no clear message. I encourage to seek the main message, but also the information necessary to transmit a key message. The introduction should provide sufficient background information to make the article intelligible with sufficient context and with links or comparison with the materials of interest in this study. What are the goals of the study?
Thanks for your constructive criticism. The introduction has been totally redirected.
The novelty of the work wasn't clearly mentioned in this section. Authors must consider reorganizing the introduction to have a more concrete understanding of what your paper will focus on and distinguish their work from other published research that can be found in the literature.
I agree, the novelty has been highlighted in the new introduction
Full words have to be written instead of abbreviations as YSTZ yttria stabilised tetragonal zirconia (YSTZ), same for 3Y-TZP.
Done.
Lines 56 and 59 3Y-TCP instead of 3Y-TZP are used.
Corrected
Line 105 and 107 the authors may use bioactive glass named RKKP bioglaze® (RKKP®).
Corrected.
MATERIALS AND METHODS
All commercial precursors used (powders and solvents: CaSiO3, ethanol; Beycostat….) with details provider and purity needs to be in the first paragraph. If synthesize the protocols need to be described. Full words have to be written instead of abbreviations : psW.
Done.
In this section, it seems that 3 kind of rods samples are elaborated MgO–MgSZ, W–TCP and W–TCP bioceramic coatings on eutectic rods what is not clear in the abstract and the introduction. In “W–TCP bioceramic coatings on eutectic rods” which rods are coated?
We have made important changes in the abstract and in the introduction to better guide the reader of the objectives of the article.
it would be beneficial to put lines from 151 to 154 in the results section. The authors should use a correct unit for “1402 ºC” .
Done
3.1. Bioactive W-TCP eutectic composite
In the tittle of 3.1. the word “composite” appears for the first time and should be mentioned earlier in the introduction and materials.
Corrected.
Lines from 184 to 196 : the paragraph should be moved in the Introduction section.
Moved.
Could the authors provide XRD analysis to define the nature of phases and their crystallinity or with amorphous behavior at 20, 100 and 150mm/h?
This crystalline and amorphous phases of this material has been described by PN de Aza, in the given references. The glass ceramic hasn't not been reported before. Unfortunately the XRD analysis is not conclusive to identify the phases, one of them does not diffract as it is vitreous and the other does not correspond to any stoichiometric phase. We have proposed the composition obtained by EDS.
Could the authors explain why : “The sample surfaces were etched with dilute acetic acid to remove the TCP phase”? And if it is part of the elaboration that could be relocated in the section MATERIALS AND METHODS.
Etching was necessary because of the lack of contrast of the phases in the FESEM images. This part has been moved to materials and methods and more details are given.
If TCP are removed, could the authors provide the nature of the 0.39% volume fraction of phases?
The fraction of phases was calculated theoretically from the eutectic composition and checked by image analysis in samples before etching.
For the eutectic rod grown at 100 mm/h, the sample surfaces were also etched with dilute acetic acid to remove the TCP phase?
It was not necessary, in this case the contrast was better and the phase analysis was easier.
Could the authors explain the difference of composition in the sample and why the sample are grown at two different rates, could they add the EDS results?
The significance of the experimental findings is not clear. Is there an interest to obtain 3 different composition and structure?
We wanted to illustrate that we can control the microstructure and hence the properties by changing the grow conditions. The most important result is that a moderate solidification speed this material can be grown in a glassy form (when the cooling rate is high enough). An this result can be used to obtain a glazed W-TCP or MgO-MgSZ eutectic. We have had a graphical abstract to clarify our purpose.
Line 266 “The samples were previously annealed for 2 hours at 600 ºC to relieve internal residual stresses” all details as this one are beneficial in the section MATERIALS AND METHODS.
Moved.
The remelting process could be explained and described with all the parameters in the section MATERIALS AND METHODS.
Moved.
The authors have to mention the conditions at which they obtained the results and add them in the figure caption.
Done.
Is the test to estimate the in vitro bioactivity could be done for this sample?
The in vitro behaviour of the eutectic glass and crystalline form has been described by us in ref. P. Velásquez et al. With this paper we tried to describe the microstructural changes. The surface properties such as bioativity, wear, wetting are under study.
3.2. MgO-MgO eutectic
The authors should correct the tittle of this section.
Corrected
Could the authors provide XRD analysis to define the nature of phases and their crystallinity?
We have described the phases of this material in a previous article (ref 22)
In Figure 5, could the authors describe the two different phases with caption. Done
Could the authors explain the difference of composition in the sample, could they add the EDS results?
Yes, we have included this analysis.
Could the authors compare and discuss the mechanical properties of the W-TCP composite and MgO-MgZ.
Yes, this is a very interesting point. We have included the comparación and discussion of mechanical properties (elastic modulus).
3.3. Surface structuring of MgO-MgSZ eutectic by laser remelting and phase dissolution
The section is not described earlier in the manuscript.
We include a mention of it in the introduction
Line 369 Could the authors edit the word lamellae.
Lamellae is correct, lamellae is the plural of lamella. We call the plate-like phase lamella.
Could the authors determine the amount or % of the Mg ions released.
Yes, the Mg released correspond to the MgO phase. We include the calculated amount in the text.
Could the authors explain the impact of presence of porosity or loose of Mg ions?
This is discussed in the text.
Could the authors provide a porosity analysis of the samples?
The porosity is 28%, the size of the pores depends on the solidification rate. We have included these aspects in the discussion.
3.4. W-TCP coatings on MgO-MgSZ rods.
Could the authors provide the in vitro behavior of the W-TCP coatings on MgO-MgSZ materials?
Could the authors describe and discuss more in this section?....
We expect the coating to behave like W-TCP glass in terms of bioactivity but this study is ongoing.
References:
we have check the references and now they are consistent
Reviewer 2 Report
The manuscript descried the development and surface modification of MgO-MgSZ eutectic ceramics prepared by LFZ method for bone restoring applications. It is interesting, enlightening and well organized. I would suggest the following minor revisions/corrections before acceptance for publication.
It is suggested to draw a schematic to show the preparation apparatus and method to enlarge the readership, for some readers may not be familiar with LFZ method.
Figs. 4-11, bars and notations are too small to read.
Lines 413-423, as mentioned as necessary, if the wettability of the surface can be assessed.
Line 136, tricalcium phosphate
Ine 149, ethanol; Beycostat, the detailed type
Line 189, In particular, such ceramics, known as Bioeutectic, has the property... have
Line 193, precipitation of Hydroxyapatite (HAp)… hydroxyapatite
Line 197, when growth by... during
Line 231, are 8 x 1027 K/s, uppe case of 27
Line 246, they a separated by a glass portion are
Line 276, toughness are less dependents on microstructure...dependent
Line 278, has been achieved with laser surface remelting…delete a
Line 285, the glass surface formation strengths the crystalline...strengthens
Lines 288, 298, at the centre an edge of the rod. and
Line 301, MgO-MgO eutectic...MgO-MgSZ or MgO-ZrO2 eutectic?
Line 322, A totally change to cellular structure occurred at solidification rates of 750 mm/h...total change, rate of
Line 402, kept in BSF for 4 weeks ...SBF
Line 404, The release of Mg ions and the inert character of zirconia does not... do not
Line 426, thermal damage generated by laser texturing or laser ablation reduce the...reduces
Line 431, is considered a positive... as (a) postive
Line 443, The power of the laser and the rod movement was adjusted...were
Line 462, (Li, Mg, Sr, Cu, Co, Mn, b, Si) to ...B
Line 491, microstructure has been grown... have
Line 502, substrate at atomic level...Only SEM analysis was carried out.
Author Response
Dear reviewer, thank you for your constructive suggestions and time spent to analize this manuscript.
All the corrections have been considered.
It is suggested to draw a schematic to show the preparation apparatus and method to enlarge the readership, for some readers may not be familiar with LFZ method.
We have prepared a graphical abstract including the description of the LFZ technique and a flow chart of the experimental for a better understanding of the sample preparation.
Figs. 4-11, bars and notations are too small to read.
The bars and notation of the figures have been improved for a better observation.
Lines 413-423, as mentioned as necessary, if the wettability of the surface can be assessed.
In the work, we present the possibilities of the laser floating zone and laser remelting to achieve a good microstructure control of the materials under study. wettability and wear characterization are under study. We have corrected the text so as not to deepen into this controversial issue.
English language and style have been improved
Reviewer 3 Report
Abstract is vague, it is not easily understandable. In first/second case what happens? Where occurs eutectic transformation?
L35: “Bioactivity is necessary to induce strong implant/bone interfaces”, this sentence is not necessarily correct. Rewrite and make it more scientific.
L38, L42: SBF test is an elementary test to evaluate bioactivity and osseogenesis process which is currently rarely used.
The aim of work in the last paragraph of introduction is unclear.
L133:”psW” means?
A lot of typos error and grammatically mistakes are in text.
L270-276: refrence
L349-350: what are these amounts? Name the parameter.
L331: how you calculate it?
L474: P in Ca/P is missing?
L481-487: this is results and discussion? Move to introduction.
I strongly recommend to the authors to describe the work in one comprehensice graphic because the work looks confusing.
I also suggest to work on the figures with some frames, insets, arrows, … for better understanding. The figures need to be cooked well.
L239: referencing in the middle of line is not norm, so please reference at the end of line.
L290: “process of nucleation and crack propagation”, rewrite to “process of crack nucleation and propagation”
Incorporate last 5 years papers of about 50%, the references are very old. These papers are suggested to strengthen literature. Homogenize the references format.
Cellulose-reinforced bioglass composite as flexible bioactive bandage to enhance bone healing - ScienceDirect
Surface modification of orthopedic implants by optimized fluorine-substituted hydroxyapatite coating: Enhancing corrosion behavior and cell function - ScienceDirect
Nanodiamonds for surface engineering of orthopedic implants: Enhanced biocompatibility in human osteosarcoma cell culture - ScienceDirect
Author Response
Dear reviewer thank you for your comments. They are very interesting and have allowed me to reconsider some aspects of the work as I comment below.
The abstract has been rewritten. We have highlighted the main novelty of the work, which is the adaptation of the zonal fusion technique to modify the surfaces from the structural and functional point of view.
L35: “Bioactivity is necessary to induce strong implant/bone interfaces”, this sentence is not necessarily correct.
L38, L42: SBF test is an elementary test to evaluate bioactivity and osseogenesis process which is currently rarely used.
The aim of work in the last paragraph of introduction is unclear.
We have rewritten the introduction to make it clearer. We have focused our work on microstructural control using the laser surface fusion technique.
L133:”psW” means? pseudo-wollastonite, it is the high temperature phase. We will call this phase wollastonite throughout the text.
A lot of typos error and grammatically mistakes are in text.
Thanks, they are corrected.
L270-276: refrence. I refer to the data in table 1.
Ok
L349-350: what are these amounts? Name the parameter.
Hardness and toughness, indicated in the text.
L331: how you calculate it?
Representing lambda (interlamellar spacing) vs. inverse-square-root of V (solidification rate) over the range of the solidification rates studied, as done in the article cited (Kennard)
L474: P in Ca/P is missing?
Corrected
L481-487: this is results and discussion? Move to introduction.
This is an article of mine. I need to refer to these results since I have not yet performed bioactivity tests on the coated materials. I would expect to obtain a result similar to that described in the cited article. I consider it part of the discussion.
I strongly recommend to the authors to describe the work in one comprehensice graphic because the work looks confusing.
You are rigth, a graphical abstract is included in the new version.
I also suggest to work on the figures with some frames, insets, arrows, … for better understanding. The figures need to be cooked well.
Your are rigth again. The pictures have been modified accordingly.
L239: referencing in the middle of line is not norm, so please reference at the end of line.
Corrected.
L290: “process of nucleation and crack propagation”, rewrite to “process of crack nucleation and propagation”.
Corrected.
Incorporate last 5 years papers of about 50%, the references are very old. These papers are suggested to strengthen literature. Homogenize the references format.
Thank you, I have taken into account your advice as you will see reflected in the text.
Cellulose-reinforced bioglass composite as flexible bioactive bandage to enhance bone healing - ScienceDirect
Surface modification of orthopedic implants by optimized fluorine-substituted hydroxyapatite coating: Enhancing corrosion behavior and cell function - ScienceDirect
Nanodiamonds for surface engineering of orthopedic implants: Enhanced biocompatibility in human osteosarcoma cell culture - ScienceDirect
Reviewer 4 Report
1 The abstract should be revised. The significances in engineering field should be highlighted. 2 The authors are suggested to explain the novelty of the paper. 3 In Section 2, the authors are suggested to add a diagram or picture of the flow chart. 4 For the analysis in Section 3, further step analysis should be added. The authors are suggested to add comments as well as the references below. They are closely related with the present research. Mechanical Systems and Signal Processing, 2023, 182: 109349. 5 The English should be improved considerately.Author Response
Dear reviewer, thank you for your helpful suggestions.
1 The abstract should be revised. The significances in engineering field should be highlighted.
2 The authors are suggested to explain the novelty of the paper.
The abstract has been rewritten according to your comments and the novelty has bee also highlighted in the introduction.
3 In Section 2, the authors are suggested to add a diagram or picture of the flow chart.
We have added a Graphical Abstract to make the objettive of the research and the metodology used more understandable.
4 For the analysis in Section 3, further step analysis should be added. The authors are suggested to add comments as well as the references below. They are closely related with the present research.
We have extended the discussion based on the results of the cited references
5 The English should be improved considerately.
The english has been improved
Round 2
Reviewer 1 Report
INTRODUCTION
Line 86-87 : “In this study, rods of wollastonite- tricalcium phosphate (W-TCP) and magnesium oxide-zirconium oxide (MgO-ZrO2) both in the eutectic composition has been grown from their melts by laser assisted directional solidification The effect of the solidification rate”
Same abbreviations and acronyms have to be written all along the manuscript
Keep using MgO-MgSZ instead of MgO-ZrO2
A dot is missing in this paragraph
MATERIALS AND METHODS
Lines 123: for “600 ºC” The authors should use a correct unit
Line 130-131: for “PVB (polyvinyl butyra” if commercial precursor, need to be described in the first paragraph with details provider and purity.
Line 132-133: “The power was adjusted to melt the coating with minimum dilution with the substrate.” Can the authors give a scale range with unit concerning the power used?
Line 158-159 : “The ion concentration was observed to be nearly equal to that of the body plasma as a SBF environment” Can the authors provide the ion nature and some values of concentration and also the pH to describe the simulated body fluid (SBF)?
Line 174: “Below this rate phosphate crystalline phases nucleate in a glassy matrix of calcium silicate producing a glass-ceramic material” Do the authors refere to peoduction of glass-ceramic below 150 and above 20mm/H? or the range are different?
Line 176 / Could the authors discuss more the SEM analyses part and explain the interest to have a certain morphology for the application
Can the authors discussed more between the two samples obtained through the SEM analyses and morphology aspect?
Can the authors discuss more the interest to produce glass ceramic or composite structure for this material? What could be the advantage or inconvenient in using the different structure?
In the caption of Figure 3. “Growth by LFZ of a glass rod of W-TCP with the eutectic composition (a). Eutectic W-TCP sample (b) grown at two different conditions” : The authors should write the exact conditions of growth.
Line 248 : The units need to be corrected.
3.2. MgO-MgSZ eutectic
Line 304-309: Could the authors the unit of the parameters used in the equation?
As previously mentioned, the authors need to discuss more about the samples obtained at the different conditions through microscope observations and strength.
3.3. Surface structuring of MgO-MgSZ eutectic by laser remelting and phase dissolution in SBF
Concerning Figures 7- 9 Could the authors provide more discussion on the results as coating thickness, interfaces, direction…. trough comparison of the different thickness, power and the two approaches of surface structuring used…
Line 369-370: Could the authors discuss more about the impact of porosity generation
3.4. W-TCP coatings on MgO-MgSZ rods
Line 409 : “Ten dips were used at a speed of 3 mm/s. No crack formation or spallation were observed on the coating surface after the sintering treatment at 1200 ⁰C during 12 h, below the melting point of 1402 ⁰C.”
This part should be in the MATERIALS AND METHODS section
The conclusion need to be improved with more details and comparison on the different approaches and materials , morphologies, composition, porosity….
Reviewer 3 Report
1. The title is suitable for a review paper, The research paper's title of a research paper should go right to the point, and specific keywords of your study need to be embedded in your title.
2. I can't see graphical abstract
3. Could the authors provide XRD analysis to define the nature of phases and their crystallinity or with amorphous behavior at 20, 100 and 150mm/h? you can not refer XRD of your research to the reference, even though its same material, you have to do again and report it.
4. Abstract: “This new method is based upon the remelting of a surface layer by using laser radiation and its versatility is demonstrated in the surface structuring of two different eutectic composites with potential application as bone substitutes”, the remelting of a surface layer?? What is A layer?
5. Add one line end of abstract you have done this work for what purpose?
6. Fig 10: disorder adjustment, pictures and scale bar.
7. Believe me I can’t get the purpose of your study. “In this study, rods of wollastonite- tricalcium phosphate (W-TCP) and magnesium oxide-zirconium oxide (MgO-ZrO2) both in the eutectic composition has been grown from their melts by laser assisted directional solidification The effect of the solidification rate on the microstructure and mechanical properties has been studied. The porosity formation due to the MgO dissolution by immersion of the MgO-MgSZ in simulated body fluid (SBF) is discussed. These materials have been used to demonstrate the possibilities of a new technique based on a modification of the laser zonal fusion technique (LFZ). In this process the surface properties of the materials are modified by melting the rod surface. This surface melting results in the formation of a glass layer on the W-TCP eutectic rods and a microstructure refinement of the surface of MgO-MgSZ eutectic rods. In addition, the adhesion of a functional glass coating on a MgO-MgSZ eutectic cylinder applying this new technique has been demonstrated.” Confusing.! Why did you do that? So?
Reviewer 4 Report
1 The abstract should be revised. The significances in engineering field should be highlighted. 2 The authors are suggested to explain the novelty of the paper. 3 In Section 2, the authors are suggested to add a diagram or picture of the flow chart of the procedure. 4. For the analysis in Section 3, further step analysis should be added. The authors are suggested to add comments as well as the references below. They are closely related with the present research.Mechanical Systems and Signal Processing, 2023, 182: 109349, Tribology International, 2023, 177: 107982. 5 The English should be improved considerately.Author Response
Please see the attachment
